# Management of Rhizosphere Microbiota and Plant Production under Drought Stress: A Comprehensive Review

**DOI:** 10.3390/plants11182437

**Published:** 2022-09-19

**Authors:** Catalina Vidal, Felipe González, Christian Santander, Rodrigo Pérez, Víctor Gallardo, Cledir Santos, Humberto Aponte, Antonieta Ruiz, Pablo Cornejo

**Affiliations:** 1Center for Research on Mycorrhizae and Agro-Environmental Sustainability (CIMYSA), Universidad de la Frontera, Temuco 4811-230, Chile; 2Doctorate in Sciences Mention Applied Cellular and Molecular Biology, Universidad de la Frontera, Temuco 4811-230, Chile; 3Environmental Engineering and Biotechnology Group, Faculty of Environmental Sciences and EU-LA-Chile Center, Universidad de Concepción, Concepción 4070-411, Chile; 4Doctorate Program in Science of Natural Resources, Universidad de la Frontera, Temuco 4811-230, Chile; 5Department of Chemical Science and Natural Resources, Universidad de La Frontera, Temuco 4811-230, Chile; 6Laboratory of Soil Microbial Ecology and Biogeochemistry, Institute of Agri-Food, Animal and Environmental Sciences (ICA3), Universidad de O’Higgins, San Fernando 3070-000, Chile; 7Escuela de Agronomía, Facultad de Ciencias Agronómicas y de los Alimentos, Pontificia Universidad Católica de Valparaíso, Quillota 2260-000, Chile

**Keywords:** arbuscular mycorrhizae, global climate change, PGP bacteria, PGP fungi, plant growth promotion, rhizosphere engineering

## Abstract

Drought generates a complex scenario worldwide in which agriculture should urgently be reframed from an integrative point of view. It includes the search for new water resources and the use of tolerant crops and genotypes, improved irrigation systems, and other less explored alternatives that are very important, such as biotechnological tools that may increase the water use efficiency. Currently, a large body of evidence highlights the role of specific strains in the main microbial rhizosphere groups (arbuscular mycorrhizal fungi, yeasts, and bacteria) on increasing the drought tolerance of their host plants through diverse plant growth-promoting (PGP) characteristics. With this background, it is possible to suggest that the joint use of distinct PGP microbes could produce positive interactions or additive beneficial effects on their host plants if their co-inoculation does not generate antagonistic responses. To date, such effects have only been partially analyzed by using single omics tools, such as genomics, metabolomics, or proteomics. However, there is a gap of information in the use of multi-omics approaches to detect interactions between PGP and host plants. This approach must be the next scale-jump in the study of the interaction of soil–plant–microorganism. In this review, we analyzed the constraints posed by drought in the framework of an increasing global demand for plant production, integrating the important role played by the rhizosphere biota as a PGP agent. Using multi-omics approaches to understand in depth the processes that occur in plants in the presence of microorganisms can allow us to modulate their combined use and drive it to increase crop yields, improving production processes to attend the growing global demand for food.

## 1. Introduction

Projections of agroclimatic models indicate a strong impact of global climate change (GCC), represented by both temperature increases of 0.5 to 2 °C by 2100 and a significant decrease in rainfall [1,2]. This change will undoubtedly promote a reconversion of agronomic practices to produce the necessary food for a growing world population, both in volume and requirements of high-quality products [3]. Regardless of the plant species cropped, agricultural production depends on a significant proportion of water being supplied mainly by rainfall, which has had a strong impact on the production of the last few seasons due to a marked mega-drought in many places around the world [1].

Under this complex scenario, the availability of water sources for irrigation is also a major structural problem, since the availability of adequate infrastructure to allow water storage is insufficient to ensure access to water during periods of higher demand [4]. Such constraints require the implementation of innovative agricultural approaches to increase resilience to climate variability, such as the incorporation of new species to be cropped. Additionally, lands where new crop species are chosen to be incorporated in must be evaluated in terms of alternatives to increase the “water use efficiency”, together with an increased plant tolerance to drought [5,6,7,8]. The above is a major challenge for global agricultural activity given the extent of the predicted climate change effects on agriculture [9].

Consequently, one of the major questions to be answered under this complex paradigm is not how to access more water in the short–medium term (that could only be answered at the infrastructural level and in the long term) but how it would be possible to keep the current yields of plant production with the scarce water availability, or even how to increase them based on the projected demand. An alternative that has generated much interest in recent years is the use of plant growth-promoting (PGP) microorganisms as inoculants (biofertilizers) in agricultural plants [6,7,10,11]. The above is based on the numerous microbial strains that have developed tolerance to the environmental stresses in which they commonly are exposed. This, together with the microbial fast growth and multiplication rate, can generate a high abundance of some desirable traits in a very short time.

Among the multiple PGP microbial groups, the arbuscular mycorrhizal (AM) fungi stand out because they establish strict mutualist symbiosis with most of the agricultural plant species, mainly characterized by providing mineral nutrition (as P) and water transport to the host plants [5,12,13]. Moreover, the development of biotechnological tools using AM fungi constitutes one of the most environmentally friendly alternatives to address the above-described constraints in a context of resilient agriculture [12,14,15,16,17]. Additionally, other free-living microorganisms such as yeast and bacteria present diverse PGP traits, which also support their use as biofertilizers. While their beneficial effects on plant growth have been extensively described, they also produce effects in plants that can be based on (or can promote) molecular, biochemical, and physiological changes that are lesser known.

Despite the increasing use of microbial inoculants from both monospecific isolates and consortia of yeast and bacteria (for instance, [7,18]), some points need to be addressed, such as: (a) the basis by which such microorganisms generate beneficial responses in plants; and (b) the multiple microbial or “ecological” interactions that may occur in the rhizosphere [19,20]. In the last case, it may not necessarily produce positive effects, being in some cases even negative [21]. Therefore, as a basis for the design of optimized bioinoculants, it is necessary to understand their degree of compatibility. This can be achieved by avoiding negative interactions such as competition or predation and promoting commensalism and cooperation [22]. Finally, the microbial functionality in generating desirable responses in the host plants will be registered as yield and food quality increases.

There are previous experiences that link, at the molecular level, the responses of plants to inoculation with certain PGP microorganisms [11,23,24]. Meanwhile, to the best of our knowledge, there are no systematized works to develop microbial consortia engineered to different hosts, including microbe–microbe interactions + PGP traits + shifts in plant mechanisms, to develop optimized biofertilizers for a specific plant species. In contrast, it is commonly concluded that the effects observed by the inoculation with a PGP microorganism in a model plant could easily be transferred to other species without a deep understanding of the physiological, biochemical, or molecular changes in the host plant (regarding the uninoculated plant) [7,10].

The above scenario generates double uncertainty: (i) whether the inoculant can be beneficial to more than one host plant species; (ii) whether the plant responses can be mechanistically equivalent between different host plants using the same inoculant. Currently, the multiple soil–microorganism–plant interactions represent an unexplored “black box”. Much less effort has been put into the validation at the field level of an inoculant’s effectiveness beyond the argued socially friendly decrease in the use of chemical inputs, as commonly advertised by marketing agricultural companies.

Against this scenario, the current “omics” tools emerge as an attainable alternative to clarify “what is happening in this black box”, allowing us to describe and predict behaviors in plants based on powerful genomic, transcriptomic, proteomic and metabolomic tools. Therefore, with this focused review, we hoped to summarize the current and updated knowledge regarding the role of microbial tools used as bioinoculants to improve the plant production. In addition, we highlight the projections based on the use of single-omics platforms and multi-omics approaches. With this, our aim was to elucidate the mechanisms that explain the improvements in yield and food quality under drought stress, as one the main sensible effects of climate change in agriculture.

## 2. The Drought as Main Constraint for Plant Production in a Global Climate Change Scenario

Agriculture meets our food demands (food security), being the main practice that contributes to the economy in many countries worldwide. However, the intensification of agriculture has also led to the degradation and exhaustion of soils; moreover, about 38% of suitable agricultural lands around the world have been degraded by inadequate management practices [25,26]. Supplying the increasing world population with sufficient food is mandatory. Because of this, food security is the most important challenge in the 21st century. However, nowadays, food security is strongly threatened by global warming [27].

Global warming and associated climate change not only affect air temperature but also influence the amount and distribution of rainfall [28]. Predictive models of GCC have shown that the frequency of precipitation events and net volumes have drastically changed during the past one hundred years. It has caused frequent and severe periods of drought in large areas around the world [29].

The agricultural sector is responsible for about 75% of the total global consumption of water [30]. However, in the wake of GCC, drought has emerged as one of the major abiotic stresses and is considered the strongest environmental stress that limits the plant growth, reducing crop productivity [31,32]. Due to the droughts, yield reductions have been reported in the order of 21 to 40%, from 1980 to 2015, for wheat (*Triticum aestivum*) and maize (*Zea mays*) productions, two of the most important crops worldwide [33]. Moreover, considering that among 80–95% of the fresh weight of plants consists of water, it is evident that drought stress leads to changes on a multidimensional level, modifying different physiological, biochemical, molecular, and morphological processes in plants [34,35,36].

The responses of plants to water stress depend on the length and severity of the water deficiency [37]. Many plants have developed mechanisms to tolerate water stress, but these mechanisms are varied and depend on the plant species. Included among them are developmental, physiological, morphological, ecological, biochemical and molecular mechanisms [38]. Mainly, the mechanisms involved in plant tolerance to drought are based on maintaining cell water homeostasis under drought conditions, allowing diminished water loss and increasing the water inlet to the plants [39].

To improve plant responses against the water stress, different approaches are currently being studied, such as traditional breeding methods, transgenic technology, and priming methods, among others. However, because of the complexity of drought effects on plants and the specific responses of plants to the water stress, each method has some problems and limitations [40,41,42].

In addition, the role of root-associated microbial communities able to improve plant drought tolerance has only been explored in recent years [43]. In this way, different studies have reported that beneficial soil microorganisms improve plant tolerance to abiotic stresses by producing a root—soil interface that directly or indirectly enhances the absorption of water and nutrients. The most important groups of microorganisms related to increased tolerance to water scarcity include the PGP rhizobacteria and AM fungi [5].

## 3. Rhizosphere Microbial Groups and Ecological Roles for Plant Production under Drought Conditions

Currently, due to the importance of the functions that soil provides, it is widely recognized that soil represents one of the main frontiers of science. The functions that soil provides strongly determine the productivity of terrestrial ecosystems, both natural and agricultural-modified ones [44,45]. The sustainable development of agriculture is mediated by biotic and abiotic factors, the rhizosphere being an environment that can be considered the common point shared by both factors. The health of the rhizosphere matrix is closely related to the state of its present PGP microbial communities [46].

In this ecological niche, factors such as the availability of oxygen, nutrients (C:N:P ratio) and water modulate a significant fraction of the total interactions in the rhizosphere [47]. In addition, in this volume of soil, it is possible to find exudates from plant roots (rhizodeposits), which are considered an essential component of the rhizosphere [48,49]. The presence or absence of rhizodeposits can significantly affect rhizosphere interactions since they fulfill signaling functions to give way to the symbiotic association between the plant and PGP microorganisms [50].

Another fundamental characteristic in the ecology of the rhizosphere is the restriction interactions of microorganisms. It has the characteristic of restricting other microorganisms from interacting with the plant (e.g., plant pathogen microorganisms). This converts the rhizosphere into a bridge of chemical communication between the plant and the PGP microorganisms for the formation of protective microbial biofilms, acting as biocontrol agents in both the defense and resistance process of the host plant [51].

The rhizosphere comprises a complex and dynamic microenvironment, with mediating characteristics in the selection of organisms that interact with the host plant. The rhizosphere can be understood as a holobiont assemblage, which is formed by the association of a host and its microbial communities [52,53]. The holobiont impacts the stability, adaptation and evolution of the organisms involved in the assemblage [54].

Under stress conditions, such as drought, salt, and poor nutrient availability, the proper functioning of the holobiont is essential for the development and growth of plants, and thus, for improving their tolerance and nutrient-obtaining capacities [55]. In summary, the ecological relationship between the rhizosphere and the characteristics of PGP microorganisms is essential for the growth, nutrition, and quality of crops, both under favorable and stress conditions. This importance is reflected even more in a context of climate change, where interactions at the rhizosphere level play a fundamental role in the health of crops [56], carbon sequestration, nutrient cycling, and rhizosphere ecosystem functioning [57,58].

Plants are closely associated with soil microorganisms both externally and internally in various ways [59]. Numerous works have demonstrated the important role played by rhizosphere microorganisms in plant growth, highlighting those with PGP capacity (Figure 1). These PGP traits have been evidenced in several studies using different rhizosphere microorganisms (Table 1 and Table 2). PGP microorganisms, in addition to benefiting plant growth, are also characterized by having the ability to produce hormones and lytic enzymes, showing a promising use as bioinoculants [60]. Some microorganisms are also involved in the production of secondary metabolites, thus promoting carbon sequestration, nitrogen fixation, and phosphorus solubilization in the rhizosphere [61,62].

PGP microorganisms (fungi, yeasts, and bacteria) are present in the soil and can contribute to maintaining or enhancing soil health, improving plant growth, inducing systemic resistance in plants, and increasing stress tolerance against different unfavorable environmental conditions, both biotic and abiotic [63]. Several works have described these PGP attributes, emphasizing mainly their ability to: (i) ensure greater nutrient availability for plants [64], (ii) stimulate changes in their root structure [13], (iii) promote the establishment and growth of plants under conditions of abiotic stress (e.g., potentially toxic elements, salinity, and drought) [5,12,65,66,67], or (iv) help with the control of phytopathogens [68].

Studies looking for beneficial microorganisms have typically used various characteristics to assess their PGP capacity, such as the microbial production of 1-aminocyclopropane-1-carboxylate (ACC) deaminase, siderophores, indole-3-acetic acid (IAA), and enzymes for the solubilization of nutrients (mainly phosphates) [6,7,14,69,70]. In addition, certain microbes (e.g., some bacteria and yeast) can potentially mitigate the phytotoxic effects by producing organic acids and extracellular polymeric substances, such as exopolysaccharides (EPSs). Additionally, AM fungi contribute with this through the production of glomalin, which is a fungal glycoprotein [18,71,72,73,74,75].

All the above beneficial traits are normally observed at rhizosphere level, with scarce reports that include the effects on host plants. The use of PGP microbes has been analyzed through effects on plants such as: (i) improvements in photosynthetic variables [7,8,76,77], (ii) biofortification of plants with essential [78,79] and beneficial nutrients [19,20], (iii) modification of antioxidant responses [66,80], and (iv) changes in secondary metabolite profiles [81,82,83], among others.

However, it should be noticed that the most recent reports highlight the increased tolerance of plants against water stress, such as those based in salinity and drought, because of direct and indirect effects by the above-described traits. In this sense, microorganisms with PGP capabilities have been shown to promote plant growth under water deficit stress conditions, suggesting that they are naturally acclimatized to the stressed environment [84].

Bacteria with PGP attributes, for instance, are key soil components able to establish beneficial associations with plants [46]. Additionally, bacteria are the most studied group of microorganisms under drought stress conditions, including the genera *Acinetobacter*, *Azospirillum*, *Azotobacter*, *Arthrobacter*, *Bacillus*, *Beijerinckia*, *Brevundimonas*, *Burkholderia*, *Clostridium*, *Delftia*, *Duganella*, *Erwinia*, *Enterobacter*, *Flavobacterium*, *Hydrogenophaga*, *Methylobacterium*, *Paenibacillus*, *Pantoea*, *Proteus*, *Providencia*, *Pseudomonas*, *Psychrobacter*, *Rhizobium*, *Serratia*, *Stenotrophomonas*, *Streptoccoccus*, and *Streptomyces* [85,86,87,88,89].

In order to enhance plant growth, the most common effect of bacteria on plants exposed to drought conditions is the improvement in the antioxidant plant responses, reducing the cell damage (mainly in membranes) and alleviating their status of stress (Table 1).

**Table 1 plants-11-02437-t001:** Recent research (2020–2022) of plant growth-promoting (PGP) bacteria and their effects on plants growing under drought stress.

Crop	Microorganism	PGP Traits Evidenced	Specific Effects	References
*Sorghum bicolor* L.	*Streptomyces laurentii* and *Penicillium* sp.	P and Zn-solubilization. Siderophores, hydrogen cyanide, NH_3_ and IAA production	+ Plant growth + Chlorophyll content + Production of osmolytes − Lipid peroxidation	[89]
*Poncirus trifoliata*	*Ochrobacetrum* sp., *Microbacterium* sp., *Enterobacter* sp., and *Enterobacter cloacae*	N-fixation, P-solubilization, ACC deaminase activity, siderophore and IAA production	+ Proline accumulation in leaves + Relative water content + Cell membrane stability index + Genes like sbP5CS2 and sbP5CS1	[59]
*Zea mays* L.	*Arthrobacter arilaitensis* and *Streptomyces pseudovenezuelae*	P-solubilization, ACC deaminase activity, IAA, siderophore, and ammonia production	+ Shoot and root lengths + Dry shoot and root weights + Chlorophyll content + Numbers of leaves	[90]
*Triticum aestivum*	*Pseudomonas azotoformans*	P-solubilization, ACC deaminase activity, EPS and IAA production. Expression of biofilm genes AdnA and FliC	+ Plant growth + Dry weight of root and shoot + Photosynthetic pigments content − CAT, SOD, and GR activity	[91]
*Glycine Max* L.	*Bacillus cereus*, *Pseudomonas otitidis*, and *Pseudomonas* sp.	P-solubilization, ACC deaminase activity, IAA and ammonia production	+ Plant growth + Stomatal density + Relative water content + Chlorophyll pigments + Sugar, protein, and proline content − MDA and H_2_O_2_	[92]
*Oryza sativa* L.	*Bacillus megaterium*, *Bacillus altitudinis*, and *Bacillus endophyticus*	ACC deaminase activity, IAA, EPS, and GA production under stress conditions	+ Plant growth + Carotenoids + Total proteins + Sugar content	[93]
*Zea mayz* L.	*Bacillus subtilis* strains (DHK and B1N1)	P-solubilization, ACC deaminase activity, IAA and siderophore production. Antagonism with *Fusarium oxysporum* and *Rhizoctonia solani*	+ Plant growth + SOD, POD, and CAT activity + Chlorophyll content + Amino acid content − ROS species	[94]
*Solanum lycopersicum*	*Streptomyces strains (IT25* and *C-2012)*	P-solubilization, siderophore production, ACC deaminase activity. Salinity tolerance (NaCl 13%)	+ Plant growth + Leaf relative water content + Proline, MDA, H_2_O_2_, and total sugar content + Gene expression of ERF1 and WRKY70 + APX activity − CAT and GPX activity	[95]
*Triticum aestivum* L.	*Pseudomonas* sp. and *Serratia marcescens*	P-solubilization, Zn-solubilization, ACC deaminase activity, IAA, EPS, siderophore, and ammonia production	+ Osmolyte accumulation + Chlorophyll and carotenoids + Zn and Fe content in grains	[96]
*Triticum aestivum* L.	*Pseudomona helmanticensis* and *Pseudomona baetica*	P-solubilization and IAA production in presence of salinity (NaCl 4%) in different drought stress	+ Soil p-available + Shoot and root dry weight + Grain yield + P uptake by shoot	[97]
*Oryza Sativa* L.	Diverse PGP microorganism	P-solubilization, siderophore, EPS, N-fixation, expression of nifH and polR genes. Drought tolerance	+ Rice seedling + Shoot length + Shoot and root fresh weight + Antioxidant capability + Proline and soluble sugar content	[98]
*Zea mays* L.	*Bacillus subtilis* and *Bacillus safensis*	P-solubilization, ACC deaminase activity, IAA, EPS, biofilm, and alginate production	+ Total chlorophyll, carotenoid, and soluble sugar − Proline accumulation − Antioxidant enzymes − ACC accumulation, ACC oxidase and ACC synthase under salt stress	[14]
*Eleusine coracana* L.	*Variovorax paradoxus*, *Ochrobactrum anthropi*, *Pseudomonas palleroniana*, *Pseudomonas fluorescens*, and *Pseudomonas palleroniana*	P-solubilization, ACC deaminase activity, IAA and siderophore production, N-fixation	+ Overall growth parameters and nutrient concentration + SOD, GPX, CAT, and APX activity + Proline, phenol, and chlorophyll − H_2_O_2_ and MDA	[99]
*Sorghum bicolor*	*Streptomyces* sp. and *Nocardiopsis* sp.	P-solubilization, ACC deaminase activity, IAA and siderophore production under drought, heat, and Cd stress	+ Plant growth and photosynthetic pigments + Translocation of Cd from root to shoot + SOD, APX, and CAT activity − MDA concentration	[100]

(+): Increase; (−): Decrease; CAT: Catalase; SOD: Superoxide dismutase; POD: Peroxidase; APX: Ascorbate perosidase; GPX: Guaiacol peroxidase; GR: Glutation reductase; GTS: Glutation transferase; PPO: polyphenoloxidase; MDA: Malondialdehyde; AsA: Ascorbic Acid; AMF: Arbuscular mycorrhizal fungi; EPS: exopolysaccharide IAA: Indole acetic acid; ABA: abscisic acid.

AM fungi are another important microbial group studied by its PGP effects. AM fungi are an obligate biotroph that depend on living root tissue for carbohydrate supply, which allows them to complete their life cycle [101]. AM fungal colonization on plant roots occurs when its hyphae penetrate the epidermis and grow extensively between and within living cortical cells, forming a very large and dynamic interface between both symbionts. It enhances plant growth and yield and also decreases the effects of several abiotic stresses [102].

This obligate symbiosis can promote the formation of stable aggregates and improve water storage in the soil through the production of glomalin, which is released in large amounts into the soil [5,75,103]. Moreover, the formation of AM symbiosis can change the efficiency of water uptake by modifying the ionic balances (Na/K) in the host plant, as well as modifying the relative expression of PIP aquaporin and ionic NHX antiporter genes under osmotic stress [5,104]. New reports using AM fungi in association with other PGP microorganisms have also shown a beneficial (synergic or additive) effect on plant growth and development when established in water-scarce environments (Table 2).

**Table 2 plants-11-02437-t002:** Recent research (2020–2022) regarding the use of arbuscular mycorrhizal fungi (AMF) with or without complements and their effects on plant under drought stress.

TCrop	AMF + Complement	Specific Effects	Reference
*Glycine max* L.	*Glomus clarum*, *Glomus mosseae*, and *Gigaspora margarita* + *Bradyrhizobium japonicum*	+ Number of nodules + Grain yield and growth + CAT and POD in seeds + Proline content + Gene expression of CAT and POD − Gene expression of P5CS, P5CR, PDH, and P5CDH	[105]
*Poncirus trifoliata*	*Funneliformis mosseae*	+ Leaf gas exchange + Soil pH, and ammonium content + H^+^-ATPase activity on shoot and roots + Regulation of H^+^-ATPase gene PtAHA2	[106]
*Poncirus trifoliata*	*Funneliformis mosseae*	+ Growth traits and leaf water potential + Gene expression of two aquaporin protein + Chlorophyll concentration	[107]
*Glycine max* L.	*Acaulospora laevis*, *Septoglomus deserticola*, and *Rhizophagus irregularis* + *Bacillus amyloliquefaciens*	+ Plant biomass + Phenol, flavonoid, glycine betaine content and GTS + GA, trans-zeatin-riboside, and IAA in seeds + ATP content and hydrolytic activities of plasma membrane − ABA	[108]
*Ephedra foliate*	*Claroideoglomus etunicatum*, *Rhizophagus intraradices*, and *Funneliformis mosseae*	+ Plant growth, chlorophyll content, nitrate and nitrite reductase activity, antioxidant activity, and ascorbic acid content + Content of proline, glucose, and total soluble protein + Sucrose phosphate synthetase activity, IAA, IBA, GA, and ABA − Glutathione level	[109]
*Nicotiana tabacum*	*Glomus versiforme* + Phosphorus supplementation	+ Osmolytes content, proline, sugars, and free amino acids + Antioxidant activities of SOD, CAT, APX, POD, and GR, and AsA and GSH content. + IAA, ABA concentrations in roots and leaves. − ROS accumulation and lipid peroxidation	[110]
*Solanum lycopersicum*	*Glomus* sp., *Sclerocystis* sp. and *Acaulospora* sp. + *Acinetobacter* sp., and *Rahnella aquatilis* + compost	+ Biomass, fruit number per plant, and fruit yield + Sugar content on shoot − PPO activity and increase of POD activity	[111]
*Camellia sinensis*	*Claroideoglomus etunicatum*	+ Plant growth and leaf water content + Antioxidant activity as SOD, CAT, GPX, and APX + Regulation of CsSODin and CsCAT genes − O_2_^−^ and MDA content	[112]
*Vaccinium corymbosum*	*Funneliformis mosseae*	+ Proteins involved in amino acid metabolism, antioxidant system, signal transduction, and photosynthesis + Carotenoid biosynthesis + Photosynthetic capacity	[113]
*Malus hupehensis*	*Rhizophagus irregularis*	+ Plant growth + Total chlorophyll content, net photosynthetic rate, stomatal conductance, and transpiration + SOD, POD, and CAT + Proline and total sugar content − Accumulation of MDA, H_2_O_2_ and O_2_^−^	[114]
*Populus cathayana*	*Rhizophagus intraradices*	+ Plant biomass, root-to-root radio, photosynthetic rate, stomatal conductance + Intercellular CO_2_ concentration and transpiration rate. + SOD, POD, soluble sugar content especially on shoot + Gene expression of PcGRF10 and PcGRF11 genes induced by AMF	[115]
*Pheonix dactylifera*	AMF consortium + plant growth-promoting rhizobacteria (PGPR) consortium	+ Plant biomass, rise of phosphorus uptake, and boosted plant-water relationship + Total soluble sugar and protein content. + Soil organic matter, phosphorus, and glomalin content − H_2_O_2_ and MDA accumulation	[116]
*Solanum lycopersicum*	*Funneliformis mosseae*, *Rhizophagus irregularis* and *Funneliformis coronatum*	− H_2_O_2_ and MDA content (especially *F. mosseae*)	[117]
*Thymus daenensis* and *Thymus bulgaris*	*Funneliformis moseae* and *Rhizophagus intraradices*	+ Root and shoot dry weight, relative water content, photosynthetic pigments, gas change, and nutritional parameters + Essential oil production + Root colonization and soil spore density − Proline, MDA, electrolyte leakage, and stomatal resistance	[118]
*Trifolium repens* L.	*Funneliformis mosseae* and *Paraglomus occultum*	+ Root total length, surface area, and volume + Leaf relative water content + SOD, CAT, POD, and ABA levels in root − MDA content	[119]

(+): Increase; (−): Decrease; CAT: Catalase; SOD: Superoxide dismutase; POD: Peroxidase; APX: Ascorbate perosidase; GPX: Guaiacol peroxidase; GR: Glutation reductase; GTS: Glutation transferase; PPO: polyphenoloxidase; MDA: Malondialdehyde; AsA: Ascorbic Acid; AMF: Arbuscular mycorrhizal fungi; IAA: Indole acetic acid; ABA: abscisic acid.

On the other hand, the yeast microbial group has also been studied for their PGP capacity. However, there are fewer studies compared to the groups described above, mainly including some works exploring its PGP attributes under water stress conditions. For instance, Silambarasan et al. [7,18] demonstrated the ability of EPS produced by yeasts to promote the formation of stable aggregates and improve the storage of water in soil.

Furthermore, recent evidence showed that yeast application can upregulate soil enzymes under drought stress conditions, which increased the nutrient content in the soil, also improving the osmotic state of roots and the activity of antioxidant enzymes in the plants treated with the strains [63]. PGP microorganisms are undoubtedly a key element for the adaptation of plants to new unfavorable environmental scenarios. Deepening studies oriented to evaluate the interspecies synergistic potential of PGP microorganisms in the current framework of climate change are required.

## 4. The “Omics” Approaches and the Development of Optimized Bioinoculants

As stated above, the use of microorganism for improved drought tolerance has shown promising results in different crops [120,121]. In this sense, rhizosphere engineering (modification in the microbial rhizosphere community by known PGP microorganisms) represents a faster and more advantageous alternative to improve drought tolerance in crops than other tools, such as genetic engineering or genetic improvement [122].

The interaction of the newly added microorganisms with the rhizosphere is the first barrier to overpass to establish a relation with the already naturally present microbiome, and in the better situation, form a mutualistic and cooperative relationship [22,123]. Then, the newly formed microbiome interacts with the plant root and may produce an increase in the physiological traits that allow the plant to confront the drought stress.

This promotes, for instance, the formation of lateral roots [124], generates increases in hormone levels (ABA, cytokinin, and gibberellin) associated with increases in the water content and the hydric status of plant organs [125,126,127], improves the expression of osmotic adjustment systems (proline), and promotes the production of antioxidant enzymes [128], among others.

However, a negative effect on the plant growth and development is also a possibility [129]. The way how this microbial community increases the tolerance to abiotic stress is not completely understood yet. This gap in information is due to the complex media where microbial communities are developing, which is the most complex compartment on the Earth’s surface [24]. In this sense, understanding the interaction among microbial communities, plants, and the biotic and abiotic factors is an essential step to improve the design of bioinoculants [130,131].

A modern approach to study this interaction is the use of “Omics” technologies, which are also being applied in microbial science (Table 3). This set of new techniques allows to integrate the information of genome, proteome, transcriptome, and metabolome and gives information about the biological changes that underlie the drought tolerance produced by advanced bioinoculants [24,132].

Metabolomics is based on tools that allow the identification of the complete metabolites synthesized by an organism and can be used for the determination of how this metabolic profile changes in different conditions, such as drought stress [139]. The usual workflow to determine a metabolic profile starts with sample acquisition, sample preparation, data acquisition, bioinformatic analysis, and biochemical interpretation [140].

The widely used techniques for separating and determining the metabolic profile from different microorganisms are based on thin-layer chromatography (TLC), column chromatography (CC), flash chromatography (FC), gas chromatography coupled to mass spectrometry (GC-MS), high-performance liquid chromatography (HPLC), liquid chromatography coupled to mass spectrometry (LC-MS and LC-MS/MS), liquid chromatography with ultraviolet, visible, fluorescence, or diode array detection (LC-UV-VIS, LC-FD, or LC-DAD), gas–liquid chromatography, and Fourier transform infrared (FTIR), near-infrared (NIR), and nuclear magnetic resonance (NMR) spectroscopies [140,141].

For example, the application of the metabolomics approach in the study of drought condition in the root of the trifoliate orange and the interaction with the AM fungus *Rhizophagus intraradices* showed a total of 88 and 17 metabolites upregulated and downregulated, respectively, also showing an improvement in the physiological status of the mycorrhized plants [142]. In the same way, the use of metabolomics tools in the application of a consortium of *Bacillus subtilis*, *B. thuringiensis*, and *B. megaterium* in chickpea showed an accumulation of riboflavin, L-asparagine, aspartate, glycerol, nicotinamide, and 3-hydroxy-3-methyglutarate under drought condition, together with the reduction in the deleterious impact on the plant status [143].

On the other hand, proteomics is the use of different technics that allows to determine the complete contents of the different proteins present in an organism under specific circumstances [144]. Proteomics analysis can be used in certain ways, such as for translational proteomics, protein–protein interaction, post-translational modification, and proteomics studies at a comparative level for the comparison of protein profiles, including drought stress [141].

The typical workflow of proteomics analysis starts with the extraction of proteins in a plant tissue of interest, digestion of proteins with specific enzymes into peptides and identification of resulting peptides by mass spectrometry, quantification of protein expression, and determination of post-translational modifications [145]. The widely used techniques used for the study of proteomes include 1D and 2D gel electrophoresis, followed by mass spectrometry with different ionization sources. Matrix-assisted laser desorption/ionization time-of-flight (MALDI-TOF MS/MS) and electrospray ionization (ESI) are the main mass spectrometry tools used in proteomics approaches [141].

With the use of proteomics approaches, it was possible to find in pepper plants inoculated with *B. licheniformis* K11 a total of 15 differential expressed proteins that confer tolerance to drought stress [145]. Similar results were observed for the inoculation of *Rhizobium leguminosarum* and *Pseudomonas putida* to *Vicia faba*, which produced changes in the proteomic profile with an improvement in the tolerance to drought stress [145].

Genomics is the study of all genes present in an organism with the respective identification of sequences, intragenic sequences, and genes structures [145]; meanwhile, transcriptomics is focused on the determination of the RNA present in the specific organs, which is highly dependent on the specific environmental condition that makes the transcriptomic highly variable [146]. In this sense, the study of genomics allows to know the potential of microorganisms to produce secondary metabolites or proteins to enhance the growth and development of drought tolerance with plant interaction.

Furthermore, transcriptomics allows to determine how this genomic potential is expressed under specific circumstances. In both cases, the starting point is the extraction of the nucleic acid, but for RNA, it is necessary to synthesize the cDNA and then sequence it in a next-generation sequencing platform [141]. The use of the integrated genomic and transcriptomic approach showed that some colonization genes such as FixL/FixK/FixJ and NodD were upregulated in the presence of beneficial microorganisms [147]. Moreover, there are several types of plant growth-promoting rhizobacteria (PGPR) with the full genome assembly, which previously have shown an improvement in the plant tolerance to drought stress, such as *B. amyloliquefaciens* [148], *Serratia plymuthica* [149], *Hartmannibacter diazotrophicus* [150], and *Rhizophagus irregularis* [151].

The omics approaches still have important bottlenecks, such as the need of different, focused, and specialized researchers; and the gap in both information and integration of data available in the worldwide databases (e.g., one-stop shop) [152]. Currently, data integration can be performed post-analysis by performing “networking” after individual analyses [153] or carrying out an integrated data analysis in parallel. This last one requires specialized tools to merge data from different platforms prior to the final interpretations of the results [132,154]. For instance, MetaboAnalyst allows for multiple integration of the metabolome, transcriptome, proteome, and genome into a wide spectrum of biological samples, including plants [155], as well as data processing and statistical analyses based on the R platform [132].

Undoubtedly, the research possibilities with the incorporation of multi-omics approaches will be a strong basis for the functional interpretation of the effects that advanced biofertilizers have on their host plants. Based on the strong progress that these approaches have had in other sciences (mainly medicine and human health), it is feasible to state that they may represent the starting point for the design of optimized biofertilizers in a wide range of plant species of agricultural and environmental interest.

## 5. Perspectives and Conclusions

As stated above, a large body of evidence highlights the role of specific microbial strains within the main rhizosphere microbial groups in conferring drought tolerance to plants. However, much less known are the physiological, molecular, and biochemical mechanisms and responses displayed by plants as a consequence of the presence and action of these PGP microorganisms.

In the case of yeasts, these microorganisms have very interesting PGP characteristics that have not been widely reported, especially regarding water stress. Nevertheless, recent studies have demonstrated their great potential as coadjutants in plant growth to cope with other abiotic stresses. Such evidence makes us presume that yeasts can be a key element with a great biotechnological value that needs to be explored as a tool to face the food shortage that promises to be accentuated with the advance of GCC.

Therefore, the next steps for the designing of biofertilizers supported by the use of multi-omics approaches could represent a significant leap in the research regarding the role of rhizosphere microbial communities in plant production under drought conditions. In this sense, the integration of omics platforms will strongly support the mechanistic understanding that underlies the use of beneficial microorganisms in plant production through rhizosphere engineering.

While it is necessary to realistically recognize that the tangible results could primarily be framed at the local level, considering particular soil conditions, rhizospheres, and crop plants, the research at the pilot scale will establish the basis for the generation and massification of optimized biofertilizers. However, some considerations must be addressed, as the development of local biological collections for the ex situ maintenance of PGP microbes able to enhance drought tolerance represents a valuable resource for testing their applicability in different crops, in line with the global advice for the maintenance of genetic resources oriented to agriculture and food production [156].

Moreover, the projection of the fundamental and mechanistic bases studied in the plant, considering different efficient rhizosphere modifications, can also be focused on different types of environmental stress, such as the low availability of nutrients, salinity, heavy metals, extreme pH values, and many other environmental and soil constraints that currently affect enormous areas of arable land surfaces worldwide.

## Figures and Tables

**Figure 1 plants-11-02437-f001:**
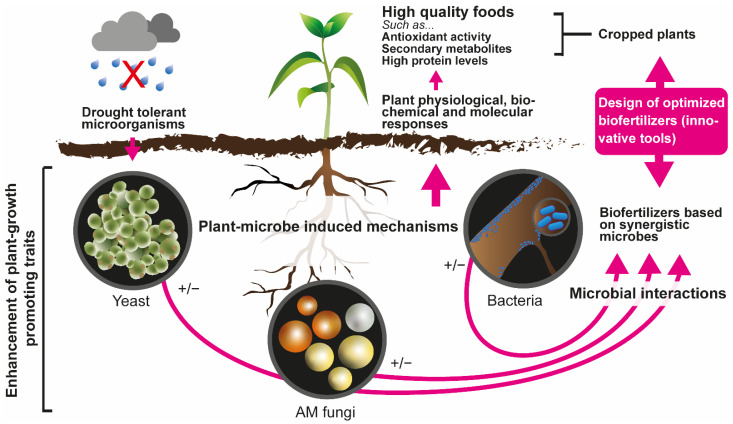
Generalized representation of the main effects of plant growth-promoting (PGP) traits on cropped plants under drought conditions as the basis for the design of optimized biofertilizers. Possible effects of PGP microorganisms on plants are shown with +/− since such effects might be positive or negative.

**Table 3 plants-11-02437-t003:** Recent research regarding the use of PGP microorganisms and their effects on plant under drought stress through omics approaches.

Methods Used	Approach	Crop	Microorganism	Crop Effects	Reference
UPLC-QTOF MS	Metabolomics	*Tritricum aestivum*	*Acremonium* *sclerotigenum*, *Sarocladium* *implicatum*	+ Proline − ABA − Lipid peroxidation − Malondialdehide − Ferulic acid	[133]
RT-QPCR	Genomic	*Oryza sativa* L.	Diverse PGP microorganism	+ Soil enzyme activities (dehydrogenase, nitrogenase, urease, and alkaline phosphatase) + Regulation of growth and stress-related genes (COX1, AP2-EREBP, GRAM, NRAMP6, NAM, GST, and DHN)	[134]
LS-MS/MS	Metabolomics	*Ananas comosus*	Consortium of *Staphylococcus* sp.	+ Indole acetic acid + ACC deaminase + Promotion of plant growth	[135]
RT-QPCR	Genomic	*Vigna mungo* L.	*Ochrobactrum pseudogrignonense* RJ12, *Pseudomonas* sp. RJ15, and *Bacillus subtilis* RJ46	+ Seed germination, root length, shoot length, and dry weight of treated plants − Regulation of ACC deaminase gene	[136]
RT-QPCR	Genomic	*Pisum sativum* L.	*Ochrobactrum pseudogrignonense* RJ12, *Pseudomonas* sp. RJ15, and *Bacillus subtilis* RJ46	+ Seed germination, root length, shoot length, and dry weight of treated plants − Regulation of ACC deaminase gene	[136]
UHPLC-HDMS	Metabolomics	*Zea mays*	*Bacillus licheniformis*, *Brevibacillus* *laterosporus*, and *Bacillus amyloliquefaciens*	+ Phenylpropanoid biosynthesis + Glycine, serine, and threonine metabolism + Tyrosine metabolism + TCA cycle metabolism	[137]
LC-ESI-QqQ-MS	Metabolomics	*Zea mays*	*Bacillus licheniformis*, *Brevibacillus* *laterosporus*, and *Bacillus amyloliquefaciens*	+ Salicylic acid + Indole-3-carboxylic acid + Glycine, cysteine, and tyrosine + Apigenin, apigetrin, and vicenin	[138]
RT-QPCR and Elisa	Genomic	*Zea mays*	*Bacillus licheniformis*, *Brevibacillus* *laterosporus*, and *Bacillus amyloliquefaciens*	+ Global DNA methylation + Regulation of PAL and FSNII gene	[138]

(+): Increase; (−): Decrease; ABA: abscisic acid; ACC: 1-aminocyclopropane-1-carboxylic acid; TCA: tricarboxylic acid.

## Data Availability

Not applicable.

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
