# Peer review of "Management of Rhizosphere Microbiota and Plant Production under Drought Stress: A Comprehensive Review"

_plants, 2022, doi:10.3390/plants11182437_

Round 1

Reviewer 1 Report

This paper reviews the role of plant growth promoting microorganisms in helping to mitigate drought stress in plants. The main takeaways are (1) PGP microorganisms, when used strategically, may support productive cropping systems even under drought and (2) understanding how to harness these PGP microorganisms to support drought tolerance in plants requires a molecular focus. The topics in the manuscript are generally well laid out and represent an important discussion. Below are my comments.

·      There are a lot of language issues throughout the manuscript. These need to be carefully addressed before publication as some of these errors may lead to misunderstanding of the authors’ meaning.

·      I noted some use of acronyms without any definition – for example GCC is used (I think to mean global climate change), but I don’t see where this is defined in the text

·      I think the authors could do some careful streamlining of the manuscript. There is a lot of repetitive text that I think can be removed and would help clarify the main points. For example, near the end of page 3 the sentence “Considering that between 80-95% of the fresh weight….” doesn’t add much value to the text and much of this information seems to be repeated in the very next sentence starting the next paragraph.

·      Figure 1 could use some work. Right now, it doesn’t seem to communicate much valuable information. The “+/-“ doesn’t really provide any information. I think it would be good to directly link figure 1 to tables 1 and 2 as these describe specific effects of PGP organisms.

·      Table 1: It looks like table 1 is only bacteria, so I recommend changing the title from “PGP microorganisms” to “PGP bacteria”

·      Yeast are listed as PGP microorganism, even included in figure 1, but there is actually very little discussion about these organisms. I think providing some conceptual discussion of the future value of studying these microorganisms might strengthen the manuscript.

Author Response

This paper reviews the role of plant growth promoting microorganisms in helping to mitigate drought stress in plants. The main takeaways are (1) PGP microorganisms, when used strategically, may support productive cropping systems even under drought and (2) understanding how to harness these PGP microorganisms to support drought tolerance in plants requires a molecular focus. The topics in the manuscript are generally well laid out and represent an important discussion. Below are my comments.

Dear reviewer, thank you very much for reading our manuscript and sending us your suggestions to improve the quality of our work. Here, we present the responses to your suggests and comments.

1)      There are a lot of language issues throughout the manuscript. These need to be carefully addressed before publication as some of these errors may lead to misunderstanding of the authors’ meaning.

R: An editorial review was performed. Please, see the new improved version.

2)     I noted some use of acronyms without any definition – for example GCC is used (I think to mean global climate change), but I don’t see where this is defined in the text.

R: The acronym GCC was defined in the first paragraph of introduction section; however, the entire manuscript was revised to ensure that there are no other undefined acronyms.

3)      I think the authors could do some careful streamlining of the manuscript. There is a lot of repetitive text that I think can be removed and would help clarify the main points. For example, near the end of page 3 the sentence “Considering that between 80-95% of the fresh weight….” doesn’t add much value to the text and much of this information seems to be repeated in the very next sentence starting the next paragraph.

R: Dear reviewer, thank you for your advice. Effectively, after a new read we found some redundant text that was taken off this new version, especially in the introduction and discussion sections (with track changes). Please, see the improved manuscript.

4)      Figure 1 could use some work. Right now, it doesn’t seem to communicate much valuable information. The “+/-“ doesn’t really provide any information. I think it would be good to directly link figure 1 to tables 1 and 2 as these describe specific effects of PGP organisms.

R: Thank you very much for your observation. We consider in some time the use of Fig. 1 as graphical abstract, but we also agree with the opinion of reviewer 3 about the simple and valuable communicative impact of Fig 1. However, we include a direct link regarding the figure with tables 1 and 2, as you can see in paragraph 4, line 1.

5)      Table 1: It looks like table 1 is only bacteria, so I recommend changing the title from “PGP microorganisms” to “PGP bacteria”.

R: Done, please, see Table 1.

6)      Yeast are listed as PGP microorganisms, even included in figure 1, but there is actually very little discussion about these organisms. I think providing some conceptual discussion of the future value of studying these microorganisms might strengthen the manuscript.

R: Thank you for your comment. We completely agree. As suggested, a conceptual discussion on yeasts was added in the last section. Please, see the new manuscript version.

Reviewer 2 Report

This paper reviewed the effect of drought on plant production in the global climate change, and then analyzed the important role of drought resistance played by PGP in the rhizosphere, furthermore, the multi-omic technologies used for analyzing the drought tolerance increase by bioinoculants were reviewed. The topic is good and interesting. There are some grammar mistakes in the manuscript:

Page 1, Abstract: Line 5, “highlights the role of specific strains into the main…” should be “highlights the role of specific strains in the main…”

Page 1, Abstract: Line 13, “the interaction soil-plan-microorganism” should be “the interaction soil-plant-microorganism”

Page 2, paragraph 2, line 2, “because establish a strict mutualist symbiosis” should be “because establishing a strict mutualist symbiosis”

Page 2, paragraph 2, line 8, “which also support its use as biofertilizers” should be “which also support their use as biofertilizers”

Page 3, Line 1, “uncertainty: first, i) will the inoculant” should be “uncertainty: i) will the inoculant”

Page 3, Line 2, “species? Second, (ii) will the plant responses be” should be “species? (ii) will the plant responses be”

The other grammar mistakes in the manuscript should be checked carefully. 

Author Response

This paper reviewed the effect of drought on plant production in the global climate change, and then analyzed the important role of drought resistance played by PGP in the rhizosphere, furthermore, the multi-omic technologies used for analyzing the drought tolerance increase by bioinoculants were reviewed. The topic is good and interesting. There are some grammar mistakes in the manuscript:

R: Dear reviewer, thank you very much for reviewing our work and for your gentle comments, All of them very helpful to improve our manuscript. All the changes you suggested were made. Additionally, the manuscript was also subjected to an additional English grammar revision.

Page 1, Abstract: Line 5, “highlights the role of specific strains into the main…” should be “highlights the role of specific strains in the main.”

R: Done.

 Page 1, Abstract: Line 13, “the interaction soil-plan-microorganism” should be “the interaction soil-plant-microorganism”.

R: Done.

Page 2, paragraph 2, line 2, “because establish a strict mutualist symbiosis” should be “because establishing a strict mutualist symbiosis”.

R: Done.

Page 2, paragraph 2, line 8, “which also support its use as biofertilizers” should be “which also support their use as biofertilizers”.

R: Done

Page 3, Line 1, “uncertainty: first, i) will the inoculant” should be “uncertainty: i) will the inoculant”. Page 3, Line 2, “species? Second, (ii) will the plant responses be” should be “species? (ii) will the plant responses be”.

R: The changes were included.

The other grammar mistakes in the manuscript should be checked carefully.

R: Done. Please, see the new version of our manuscript.

Reviewer 3 Report

The current version of the review entitled: " Management of rhizosphere microbiota and plant production

under drought stress: a comprehensive review" is well presented and structured. All the studies about the topic have been analyzed and interpreted to synthesize the recent advances about the topic. Even the figure has a great communicative impact.

The topic turns out to be of great scientific interest and considering these premises, I recommend the paper for publication in its present form.

Author Response

The current version of the review entitled " Management of rhizosphere microbiota and plant production under drought stress: a comprehensive review" is well presented and structured. All the studies about the topic have been analyzed and interpreted to synthesize the recent advances about the topic. Even the figure has a great communicative impact.

The topic turns out to be of great scientific interest and considering these premises, I recommend the paper for publication in its present form.

R: Dear reviewer, we greatly appreciate your kind comments. We also appreciate your time and dedication to review our manuscript. Just to let you know that aimed to improve the quality of our manuscript,  the document was also English edited and reviewed.